# Exploring Traffic Congestion on Urban Expressways Considering Drivers' Unreasonable Behavior at Merge/Diverge Sections in China

**Kejun Long [1], Qin Lin [2], Jian Gu [1,3,*], Wei Wu [2] and Lee D. Han [4]**

1   Hunan Key Laboratory of Smart Roadway and Cooperative Vehicle-Infrastructure Systems,
    Changsha University of Science & Technology, Changsha 410004, China; longkejun@csust.edu.cn
2   School of Traffic and Transportation Engineering, Changsha University of Science & Technology,
    Changsha 410004, China; linqin@stu.csust.edu.cn (Q.L.); jiaotongweiwu@csust.edu.cn (W.W.)
3   Engineering Research Center of Catastrophic Prophylaxis and Treatment of Road & Traffic Safety of Ministry
    of Education, Changsha University of Science & Technology, Changsha 410114, China
4   Department of Civil and Environmental Engineering, The University of Tennessee, Knoxville, TN 37996,
    USA; lhan@utk.edu
*   Correspondence: gujian@csust.edu.cn; Tel.: +86-731-8525-8575

**Abstract:** The mechanisms of traffic congestion generation are more than complicated, due to complex geometric road designs and complicated driving behavior at urban expressways in China. We employ a cell transmission model (CTM) to simulate the traffic flow spatiotemporal evolution process along the expressway, and reveal the characteristics of traffic congestion occurrence and propagation. Here, we apply the variable-length-cell CTM to adapt the complicated road geometry and configuration, and propose the merge section CTM considering drivers' mandatory lane-changing and other unreasonable behavior at the on-ramp merge section, and propose the diverge section CTM considering queue length end extending the expressway mainline to generate a dynamic bottleneck at the diverge section. In the new improved CTM model, we introduce merge ratio and diverge ratio to describe the effect of driver behavior at the merge and diverge section. We conduct simulations on the real urban expressway in China, with results showing that the merge section and diverge section are the original location of expressway traffic congestion generation, and on/off-ramp traffic flow has a great effect on the expressway mainline operation. When on-ramp traffic volume increases by 40%, the merge section delay increases by 35%, and when off-ramp capacity increases by 100 veh/hr, the diverge section delay decreases about by 10%, which proves the strong interaction between expressway and adjacent road networks. Our results provide the underlying insights of traffic congestion mechanism in urban expressway in China, which can be used to better understand and manage this issue.

**Keywords:** urban expressway; cell transmission model; mandatory merge; merge section; diverge section

## 1. Introduction

To alleviate traffic congestion, expressways were built in many Chinese cities, including Beijing, Shanghai, Guangzhou, and Tianjin. Compared with arterial roadways, expressways can provide continuous and rapid movement by grade separation and fully enclosed routes. Due to its key role in the whole road network, expressway traffic congestion has attracted the broad concern of travelers and traffic managers. According to the 2017 annual traffic analysis report of China's major cities, the traffic congestion index of the average road in Beijing was 1.95 in rush hour, while the

expressway reached 2.5. In Jinan, the traffic congestion index of averaged road was 1.95 during rush hour, while the expressway reached 3.0. In order to scientifically formulate an urban expressway traffic congestion management plan, it is important to deeply reveal and accurately simulate the generation and propagation mechanism of traffic congestion.

Generally, the main features of a Chinese urban expressway include:

- Short distance between on/off-ramps, resulting in numerous merging and weaving sections along the expressway mainline. According to the 2017 annual traffic analysis report of China's major cities [1], the average distance between the on/off-ramps of expressways are 850, 850, and 890 m in Beijing, Xiamen, and Dalian, respectively.
- Drivers do not strictly follow the "mainline priority" in the merge/weave sections along the urban expressway. Vehicles from on-ramp often enter the expressway after mandatory lane changing, forcing vehicles on the mainline to decelerate intensely to avoid collision.
- Strong interaction between expressways and their adjacent road network. On-ramps and off-ramps are directly connected with the road network. Due to the short ramp length, traffic flow disturbance and congestion that occurs in the local road network may quickly spread to the expressway mainline, and vice versa.
- Large traffic volume on the expressway. Although the expressway mileage reached about 9.0% of the whole road network in Beijing, the ratio of traffic volume transported by expressway reached about 34.3%.
- Traffic congestion on expressway usually occurs in merge and weaving areas. Unreasonable lane changing behavior, forced changing lane, and unreasonable deceleration behavior are the fundamental reasons for traffic congestion on expressways [2,3].

In conclusion, it is hard to solve the urban expressway traffic congestion problems by simply relying on the road construction in China. Therefore, in order to alleviate the traffic congestion, we should develop certain methods, from traffic control and management aspects, to guide drivers' behavior. This paper will supply a theoretical foundation for traffic control and management by exploring traffic congestion mechanisms on urban expressway considering drivers' behavior.

## 2. Literature Review

Over the past decades, considerable research has been conducted on the expressway congestion issue, which includes traffic flow model, traffic flow simulation, and traffic congestion detection. These fruitful studies construct the foundation for freeway congestion mechanism analysis. However, these fruits fail to simulate traffic operation on a Chinese urban expressway precisely, without considering the particular driving behavior in China.

### 2.1. Traffic Congestion Detection

Traditional traffic detection technologies include video detection and inductive loop detection. In recent years, new detection technologies appeared continuously. Nantes A. et al. [4] collected expressway traffic data using a Bluetooth device installed in vehicles, and proposed a new traffic state prediction model. Dongre et al. [5] used vehicle to vehicle (V2V) and vehicle to infrastructure (V2I) communication to provide traffic information for drivers and the traffic control system. Yuan et al. [6] proposed an expressway traffic congestion detection and control strategy based on VANET, and conducted simulations using TRANSMODELER. Néstor et al. [7] started from the idea of big data analysis, and proposed a road network traffic state detection method which combined in-vehicle information with roadside facilities data.

### 2.2. Expressway Traffic Flow Model and Simulation

Traffic flow model is the basis of traffic congestion mechanism analysis and traffic flow simulation. At present, there are two types of traffic flow model: the macroscopic model and the microscopic model.

- Macroscopic traffic flow model

   The earliest macroscopic traffic flow theory is the LWR model proposed by Lighthill, Whitham, and Richards, in which traffic flow was regarded as a continuous and compressible fluid. Then, several other macroscopic models, such as the hydrodynamic and kinematic model, were proposed successively [8]. Chen et al. [9] researched the expressway bottleneck, and proposed a traffic flow collapse probability model based on the distribution of the time headway and space headway, which revealed the inherent complexity and stochastic evolution mechanism of expressway traffic flow. Scholars et al. [10] conducted simulation on urban roads using cell transmission model (CTM), and revealed a traffic congestion evolution mechanism in the road network. The results showed that CTM could accurately predict the occurrence time and spreading range of traffic congestion. Yuan et al. [11] combined the Kalman filter and LWR model to estimate the traffic state. Wei et al. [12] proposed a large-capacity CTM, and applied it to optimize the aviation routes. Shi et al. [13] tested hysteresis loops of road detectors using macroscopic fundamental diagram, the results of which would provide a theoretical basis for predicting traffic states. Yan et al. [14] used the non-oscillating center format algorithm to solve the LWR model on non-uniform roads. Yang et al. [15] established an improved CTM based on the hysteresis phenomenon, and discussed the effect of irregular road geometric features on expressway traffic flow transmission.

- Microscopic traffic flow model

   The microscopic traffic model takes the single vehicle as the research object, and analyzes the traffic flow characteristics. At present, the main microscopic models include car-following model, lane change model, and cellular automata model. Sun et al. [16] researched on road bottlenecks and simulated on-ramp traffic flow based on the car-following theory. Chen and Laval [17] researched the single bottleneck and proposed a new car-following model based on the stop and go wave data, which revealed the effect of driver behavior on traffic oscillations. Mei et al. [18] simulated on-ramp lane-changing behavior, and found that it would take less time to generate congestion on expressways when on-ramp traffic volume increased. Cellular automaton (CA) is another microscopic traffic model, proposed by Von Neumann in the 1940s. Cellular automaton can simulate the evolution of traffic flow in discrete time. Some researchers applied CA to analyze traffic congestion formation at the merge section.

   To sum up, the macroscopic model focuses on traffic flow integrity and dynamic behavior. As a result, macroscopic model has a deficiency in that it fails to accurately describe the actual traffic flow for some traffic phenomena. For example, it cannot simulate the effect of microscopic behavior, such as mandatory merges and car-following, on traffic congestion. On the other side, although microscopic model can capture drivers' microscopic behavior, it has shortcomings. In the real world, each individual has entirely different driving behavior, therefore, it is almost impossible to construct one model to simulate all different drivers' behavior under different conditions and, in addition, calibrating the microscopic model is extremely time-consuming and laborious. Hence, such a kind of model is needed, which could describe the traffic flow state while considering the drivers' behavior.

   The main goal of this research is to reveal the mechanism, and simulate the evolution process of traffic congestion on urban expressway in China. Firstly, cell transmission model (CTM) can simulate the evolution process of traffic flow in spatiotemporal dimension, and adapt to dynamic modeling for traffic flow. Secondly, CTM has obvious superiority in the oversaturated traffic simulation [19]. In addition, the accuracy of CTM analysis can meet the requirement of traffic flow analysis and traffic management evaluation. To this end, CTM was selected as the analyzing tool.

   In addition, considering the particular features listed above on a Chinese urban expressway, CTM is improved in this paper. First, a "variable length" cell model was selected to treat expressways' complicated geometry and configuration. Then, the merge and diverge section CTM models were proposed, which can consider the effect of driving behavior, such as forced merges and queue spill-out from adjacent road networks. Accordingly, the formula of cellular flow transmission was deduced,

and a new improved CTM was established. In the case study, one elevated expressway in Changsha China was selected, and the traffic flow was modeled and simulated by using the improved CTMs. Based on the simulation results, the paper revealed the effect of on-ramp traffic flow input on mainline traffic congestion and the interaction between the adjacent local street capacity and the expressway diverge section congestion. The research results will provide a theoretical basis for real-time traffic congestion prediction, traffic management control, and traffic guidance on urban expressways.

## 3. Methodology

### 3.1. Basic Idea

CTM was originally put forward by Daganzo in 1993 [20], then widely used to study the evolution of dynamic traffic flow in road networks [21], and its core idea is to divide the road into multiple cells of equal length, and to describe the traffic flow propagation by using the transmission process of vehicle numbers between adjacent cells. Due to the discrete time and space of the CTM model and its simple structure, it is widely used in the process of traffic flow simulation research. However, the traditional CTM has obvious shortcomings when applied to simulate traffic operation on urban expressway in China, which include the following.

Firstly, the traditional CTM assumes that all cells have the same length. Since the geometric features of actual urban expressways are complex, and simulated roads may not be exact integer multiples of the uniform cell length, the traditional CTM cannot accurately describe urban roads. To solve this problem, we introduced variable-length-cell in the simulation.

Secondly, the traditional CTM strictly follows the "mainline priority" operating rule, while there are many common practical phenomena that do not match the rules, such as "mandatory merge", unreasonable lane-changing, and other unreasonable driving behavior, on Chinese expressways.

Last but not least, the traditional CTM assumes that the terminal cell capacity is infinite, while the expressway off-ramps connect with local roads, and off-ramp queue length often extends the expressway mainline upstream, due to the unmatched capacity of local streets.

To this end, considering the situations above, this paper improved the CTM in the following aspects:

- Cell length parameters were introduced to CTM to accurately describe the complex and variable geometric shapes of the expressway.
- The merge section is divided into three cells: the upstream mainline cell, on-ramp cell, and downstream mainline cell. On domestic roads, there is no clear rule that drivers must obey the "mainline vehicle priority," so forced merges and crossing multi-lane merges are general phenomena which have a great effect on the expressway mainline. Therefore, the merge ratio was introduced to improve the traditional CTM, in which the "forced merge" behavior can be considered in the merge area.
- The diverge section is divided into three cells: the upstream mainline cell, off-ramp cell, and downstream mainline cell. On Chinese expressways, congestion often occurred in the expressway diverge section site. Due to local streets, the remaining capacity is limited, and the off-ramp queue often extends the mainline rapidly to congest the expressway. In order to describe such traffic operation features, we introduced the capacity parameters for the off-ramp. Compared with the traditional CTM, the capacity of the off-ramp cell is no longer infinite; instead, when the off-ramp traffic volume is larger than the capacity of the local street, congestion will generate on the off-ramp, and spread to the mainline.

In addition, this study makes the following assumptions:

- There is, at most, one on-ramp or off-ramp in a single cell.
- There is a single cell including an on-ramp (off-ramp) in the start (end) position of the cell series.
- The basic road section is formed by a single cell without an on-ramp (off-ramp).
- There is the same number of traffic lanes in a single cell.

### 3.2. Improved CTM

The CTM assumes that the volume–density relationship follows the trapezoidal fundamental diagram, illustrated in Figure 1. In the case of low density, free flow velocity is a constant *vf*. In the case of heavy density or traffic congestion, the shock wave speed is also a constant $\omega$, and generally satisfies *vf* > $\omega$. The traffic flow transmission relation is shown as Equation (1):

$$q = \left\{ v_f k, Q_{\max}, \omega(k_{jam} - k) \right\} \tag{1}$$

Here, *q* is traffic volume; *k* is traffic flow density; $Q_{max}$ is the capacity of the road section, namely, the maximum traffic flow rate of the road section; $k_{jam}$ is the congestion density; $v_f$ is the free flow speed; $\omega$ is the shock wave speed when congestion occurs; and $k_A$ and $k_B$ are the corresponding minimum and maximum density when the traffic volume is maximum in the trapezoidal fundamental diagram.

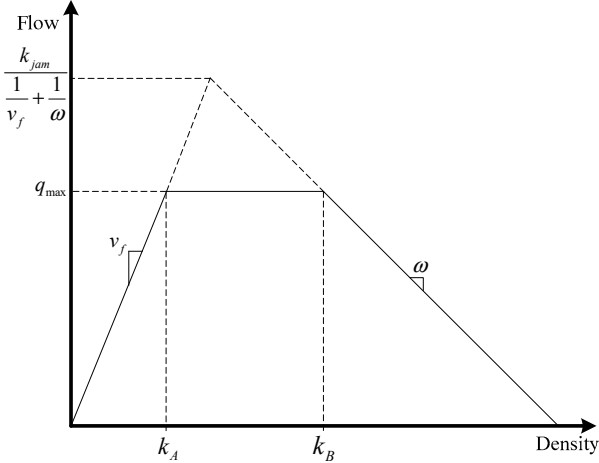

**Figure 1.** Trapezoidal fundamental diagram.

Cell length parameter $l_i$ is introduced in the basic CTM, as shown in Figure 2, $L_c$ is the standardized cell length of traditional CTM. The length of cells can be an inequality in the improved CTM, and must be greater than $L_c$, that the vehicle travels at the free flow speed within unit time. Clearly, we can set $L_c$ according to the simulating time step length.

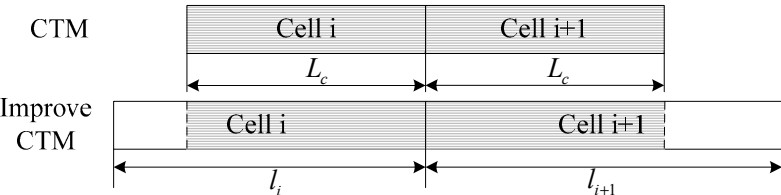

**Figure 2.** Configuration of the traditional and improved cell transmission model (CTM).

According to the traffic flow transmission relationship among cells in improved CTM, within the unit time step $\sigma$, the number of vehicles leaving cell *i* is only related to the length $L_c$ in the latter shade part in Figure 1. Similarly, the number of vehicles received by downstream cell *i* + 1 only includes the number of vehicles within $L_c$ in cell *i*. Therefore, the actual transmission number that flows into cell *i* at time *t* is related to $l_i$ and $L_c$, as shown in Equation (2):

$$y_{i+1}(t) = \min\left\{ \frac{L_c}{l_i} n_i(t), Q_{i+1}(t), \frac{L_c \omega}{l_{i+1} v_f}[N_{i+1}(t) - n_{i+1}(t)] \right\} \tag{2}$$

Here, $n_i(t)$ is the quantity of vehicles in cell *i* at time *t*; $y_i(t)$ is the actual transmission quantity of cell *i* to cell *i* + 1; $N_i(t)$ is the max quantity of vehicles in cell *i* at time *t*; $Q_i(t)$ is the max actual transmission quantity cell *i* at time *t*; $\omega$ is the shock wave speed; $v_f$ is the free flow speed.

The CTM of the expressway mainly includes three types of cell models: basic segment, merge section, diverge section, as shown in Figure 3.

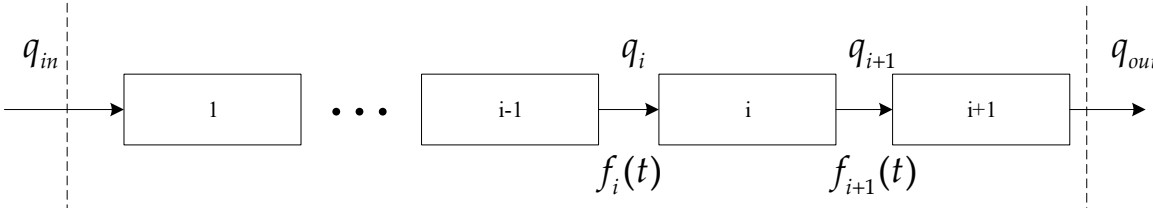

**(a) Schematic Diagram of the Basic Segment CTM Structure**

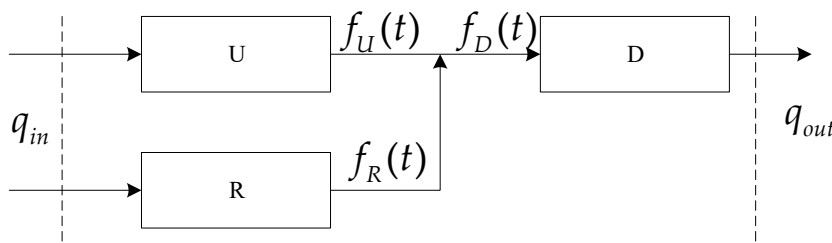

**(b) Schematic Diagram of the CTM Structure in the Merge Section**

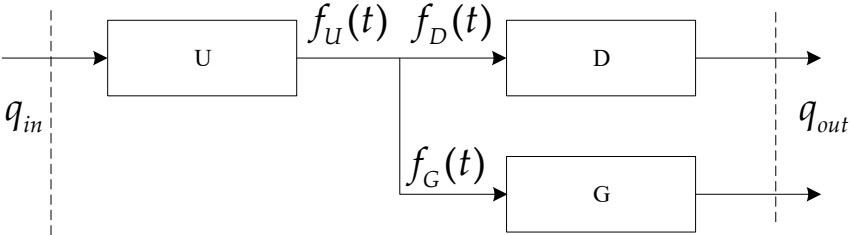

**(c) Schematic Diagram of the CTM Structure in the Diverge Section**

Legend

| U | mainline upstream cell | | D | mainline downstream cell |
|---|------------------------|---|---|--------------------------|
| R | on-ramp cell | | G | off-ramp cell |

**Figure 3.** Schematic configuration of CTM. (* Cited from Daganzo [20]).

### 3.2.1. Basic Road Segment Cell Model

From Figure 3a, according to the continuous conservation principle of inter-cellular traffic flow in LWR theory, we discretize its time and space, and establish the cell transmission model for the basic road section, as follows:

$$n_i(t) = n_i(t-1) + y_{i-1}(t-1) - y_i(t-1) \tag{3}$$

$$y_i(t) = \min\left\{ n_{i-1}(t), Q_i(t), \omega/v_f [N_i(t) - n_i(t)] \right\} \tag{4}$$

### 3.2.2. Merge Section Cell Model

Figure 3b illustrates the schematic configuration of merge section on the expressway. Merge section contains three cell parts: cell *U* is expressway mainline upstream, cell *R* is on-ramp, and cell *D*

is expressway mainline downstream. Within $t$ time interval, traffic flow output of cell $U$ is $f_U(t)$, the receiving flow is $A_U(t)$, and the sending flow is $S_U(t)$; while traffic flow output of cell $R$ is $f_R(t)$, the receiving flow is $A_R(t)$, and the sending flow is $S_R(t)$; and traffic flow of cell $D$ is $f_D(t)$, the receiving flow is $A_D(t)$, and the sending flow is $S_D(t)$. According to the definition of cell receiving flow and sending flow, the merge area meets the following restrictions:

$$\begin{cases} f_U(t) \leq S_U(t) \\ f_R(t) \leq S_R(t) \\ f_D(t) = f_U(t) + f_R(t) \leq A_D(t) \end{cases} \tag{5}$$

Considering whether the receiving flow of downstream cell $D$ is sufficient or not, there are two cases for merge cell model.

Case 1: When $A_D(t) \geq S_U(t) + S_R(t)$

$$\begin{cases} f_U(t) = S_U(t) \\ f_R(t) = S_R(t) \\ f_D(t) = f_U(t) + f_R(t) \end{cases} \tag{6}$$

Case 2: When $A_D(t) < S_U(t) + S_R(t)$, assume that merge ratio of ramp cell $R$ is $r_R(t)$, denoting the ratio of ramp volume entered mainline to mainline volume downstream, then,

$$\begin{cases} f_U(t) = \min\{S_U(t), A_D(t) - S_U(t), A_D(t)[1 - r_R(t)]\} \\ f_R(t) = \min\{S_R(t), A_D(t) - S_R(t), A_D(t)r_R(t)\} \\ f_D(t) = f_U(t) + f_R(t) = A_D(t) \end{cases} . \tag{7}$$

In case 2, when the traffic volume from expressway mainline and on-ramp are both extremely large, the receiving flow of downstream cell $D$ is insufficient, and it cannot accommodate the demand of upstream cell $U$ and ramp cell $R$. Then, the merge area will experience congestion, and the transmission of traffic flow is determined by downstream cells. In the traditional CTM model, vehicles entering the merge section should obey the "mainline priority" rule, in which ramp vehicles must give way to mainline vehicles. However, in the real-world vehicles will not obey this "mainline priority" rule. There are numerous forced merges and unreasonable lane-changing in the merge sections, resulting in congestion frequently at merge area. To represent this character, merge ratio $r_R(t)$ is introduced omto CTM model. In this paper, the merge ratio $r_R(t)$ is set as a certain value, which can express the fact that ramp vehicles do not fully obey the "mainline priority" rule. Equations (3), (6), and (7) constitute the fundamental model of the merge area based on the improved CTM.

3.2.3. Diverge Section Cell Model

Figure 3c illustrates the schematic diagram of the diverge section. The upstream mainline cell is $U$, off-ramp cell is $G$, traffic flow output of ramp cell $G$ is $f_G(t)$, the receiving flow is $A_G(t)$, and the sending flow is $S_G(t)$. Therefore, the off-ramp cell and mainline cell in the diverge section should meet the following restrictions:

$$\begin{cases} f_U(t) = f_G(t) + f_D(t) \leq S_U(t) \\ f_D(t) \leq S_D(t) \\ f_G(t) \leq S_G(t) \end{cases} . \tag{8}$$

Introducing another variable diverge ratio, $r_G(t)$, to denote the ratio of off-ramp volume to the mainline volume, and off-ramp capacity as $C_G$, then

$$\begin{cases} f_U(t) = \min\{S_U(t), A_G(t)/r_G(t), A_D(t)/[1 - r_G(t)]\} \\ f_G(t) = \min\{A_G(t), f_U(t) \cdot r_G(t), C_{G,i}/r_G(t)\} \\ f_D(t) = f_U(t) \cdot [1 - r_G(t)] \end{cases} . \tag{9}$$

In the normal case, the off-ramp capacity, $C_G$, is mostly determined by its connecting local street or local intersection. When the traffic demand exceeds the off-ramp capacity, the vehicle in the off-ramp will get stuck in the off-ramp cell, and cause traffic congestion, and even spread to the mainline. Formulas (3) and (9) constitute the basic diverge section model, based on the improved CTM.

## 4. Simulation

### 4.1. Road General Condition

To accurately describe and analyze the characteristics of urban expressway traffic flow and congestion, the Wanjiali Expressway in Changsha was selected in this study. Wanjiali Expressway is a two-way, six-lane urban elevated expressway. The total length is 16.6 km; the speed limit is 80 km/h, without traffic signal lamps; and the ramp spacing is non-uniform. Figure 4 shows the general schematic diagram of the Wanjiali Expressway:

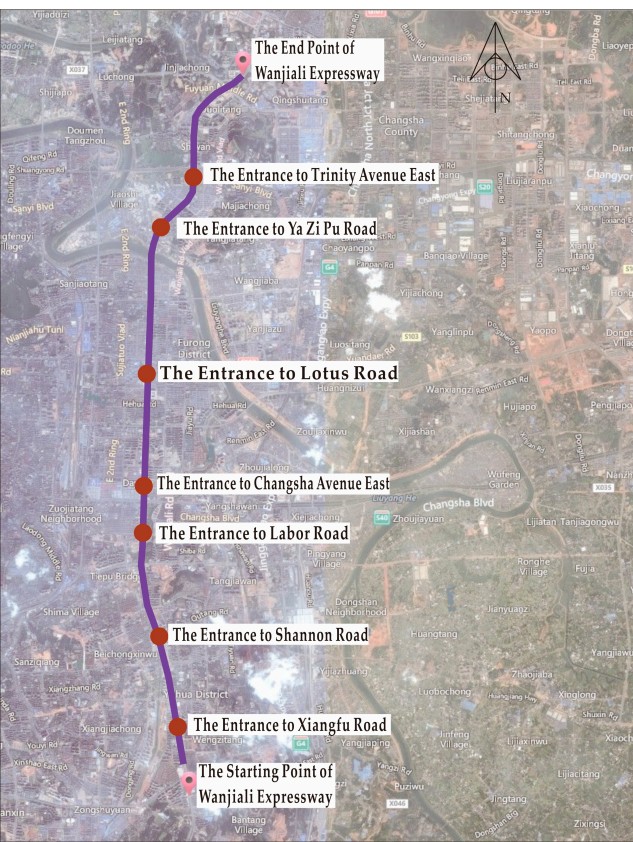

**Figure 4.** Layout of the Wanjiali Expressway.

According to the improved CTM, The Wanjiali Expressway is divided into 28 variable length cells, from south to north. As shown in Figure 5, it contains 17 basic segment cells, 5 merge section cells, 4 weaving section cells, 2 diverge section cells, and 13 ramp cells, including 7 on-ramp cells and 6 off-ramp cells. The improved CTM is used to simulate and analyze the expressway traffic flow.

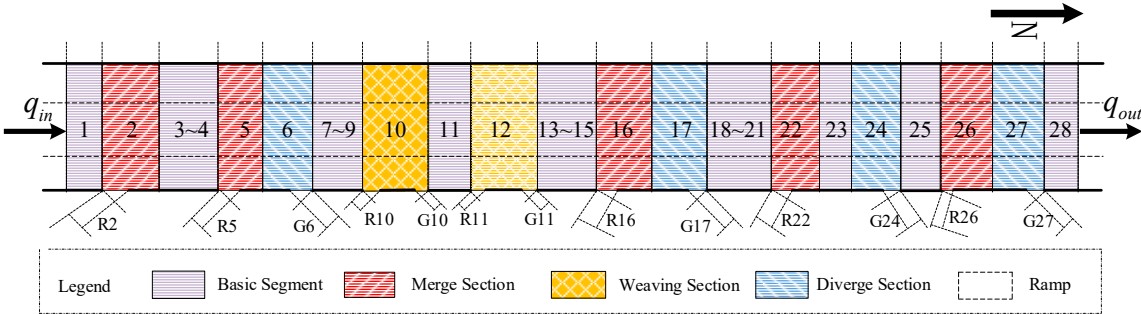

**Figure 5.** Schematic diagram of cell division.

### 4.2. Parameter Calibration

The core of CTM model calibration is the parameters of the flow speed model. In this paper, we analyzed the traffic data of the Wanjiali Elevated Expressway. We collected traffic flow and traffic speed data at the merge section of the on-ramp in Renmin Road, where the time interval of data acquisition is five minutes from 00:00 to 24:00, and conducted curve regression analysis based on a flow velocity model. The calibration is operated using SPSS software packet. Fitted with the data, the result is illustrated in Figure 6 and Table 1:

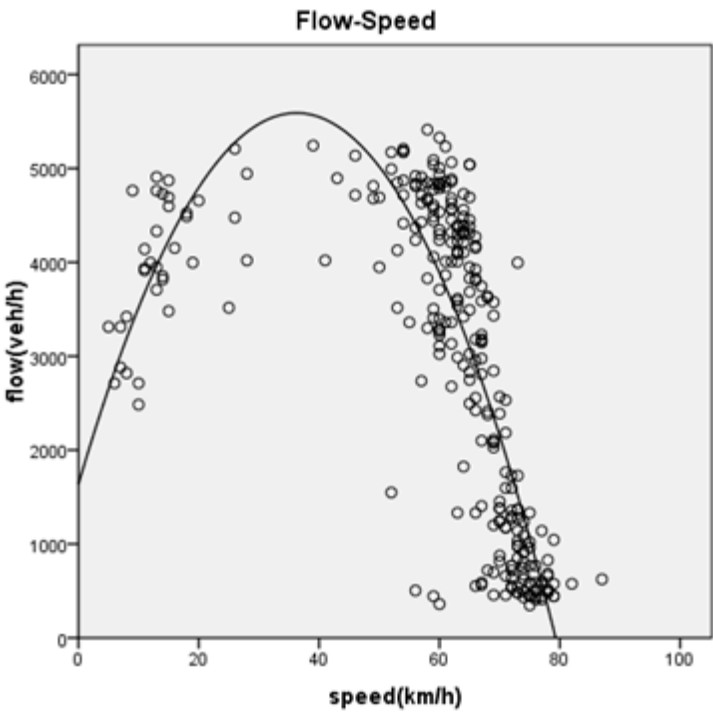

**Figure 6.** Flow speed data fitting.

**Table 1.** Result of flow speed regression analysis by SPSS.

| | Coefficient | | | | |
| --- | --- | --- | --- | --- | --- |
| | **Not Standardized Coefficient** | | **Standardized Coefficient** | **T** | **Sig.** |
| | **B** | **Standardized Error** | **Beta** | | |
| speed | 218.310 | 14.141 | 2.445 | 15.438 | 0.000 |
| speed ˆ 2 | −3.013 | 0.158 | −3.017 | −19.053 | 0.000 |
| constant term | 1635.770 | 282.139 | | 5.798 | 0.000 |

As shown in Table 1, the significant value, 0.000, is within the confidence interval, the flow speed relationship follows a quadratic curve quite well, and flow speed model can be written as

$$q = -3.013v^2 + 218.310v + 1635.77. \tag{10}$$

When the traffic flow is close to zero, the speed is close to 80 km/h, and traffic is in free flow, which is consistent with the real world.

After calibration, the main parameters of the simulation model are as shown in Table 2.

**Table 2.** Value of calibrated model parameters.

| Parameter | Unit | Value |
| --- | --- | --- |
| Simulation Time Length, $T$ | hour | 24 |
| Time Step interval, $\sigma$ | second | 10 |
| Mainline Capacity, $Q_{max}$ | veh/hr | 5400 |
| On-Ramp Capacity, $Qr_{max}$ | veh/hr | 1200 |
| Free Flow Speed, $v_f$ | km/h | 75 |
| Traffic Back Propagation Speed, $\omega$ | km/h | 25 |
| Jam Density, $k_{jam}$ | veh/(km × ln) | 122 |
| On-Ramp Merge Ratio, $r_R$ | — | 0.3 |
| Off-Ramp Diverge Ratio, $r_G$ | — | 0.1 |
| Off-Ramp Capacity Local Street, $C_{Gi}$ ($i$ = 6, 10, 12, 17, 24, 27) | veh/hr | (1000, 1200, 1100, 800, 1200, 2000) |

## 5. Simulation Results

### 5.1. Density/Delay Spatiotemporal Distribution

We collected traffic volume data and speed data from Wanjiali Elevated Expressway, in which traffic volume detectors include seven on-ramp detectors and three expressway mainline detectors, and speed detectors are located on the mainline near an on-ramp merge section. Traffic data on Wednesday, from 0:00 to 24:00, 13 December 2017, was collected for simulation as weekday traffic flow data. Traffic data on Saturday between 0:00 and 24:00, 16 December 2017, was collected for simulation as weekend traffic flow data. We conducted simulations on the weekday and weekend using the improved CTM, and obtained the detailed results which include cell density, delay, and the spatiotemporal evolution diagram of the expressway traffic flow, as shown in Figure 7.

Figure 7 mainly shows the density–time evolution at merge section, diverge section, and weaving section, in which Figure 7a represents weekday, and Figure 7b represents weekend. From Figure 7a, the traffic density rose at around 06:00 (simulation time 2200 s), and reached peak value at 08:00 (simulation time 3000 s), and began to decrease gradually at around 19:00 (simulation time 7000 s). From Figure 7b, traffic density rose at 06:00 and reached peak at 08:30, and gradually decreased at around 21:30. The simulation results are consistent with the real operation on Wanjiali Expressway.

The traffic operating index of all cells in Wanjiali Expressway on weekday and weekend was represented in Figure 8. In the wave curve, apex and peak locate at the merge section, diverge section, or weaving section. The average density and delay on weekends are larger than weekdays. In addition, according to the diverge section density of the off-ramp, we found that the smaller the capacity of the local street connecting to the off-ramp, the larger the cell delay in the diverge area.

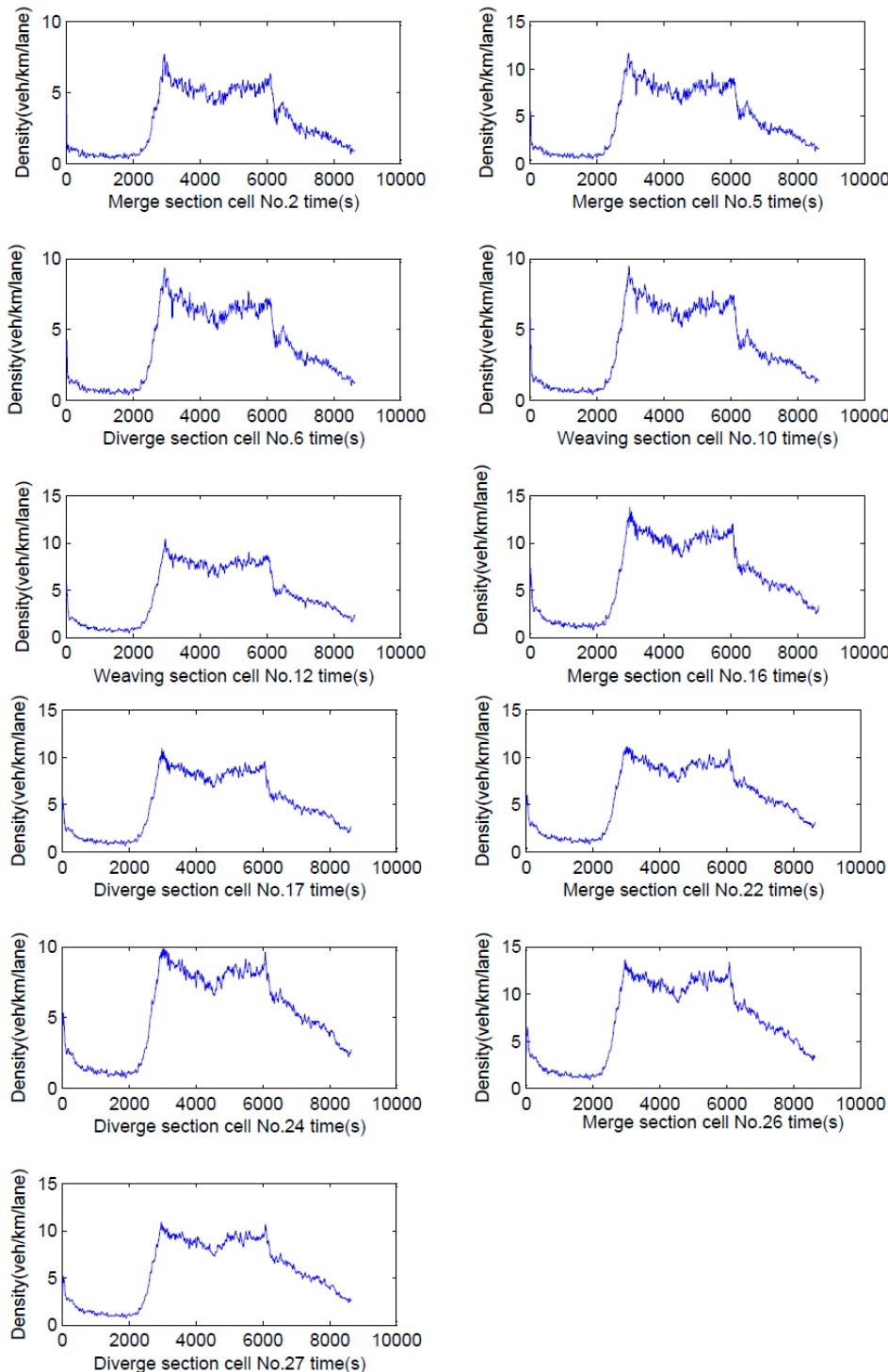

（a）Diagram of merge/diverge/weaving section density on weekdays

**Figure 7.** *Cont.*

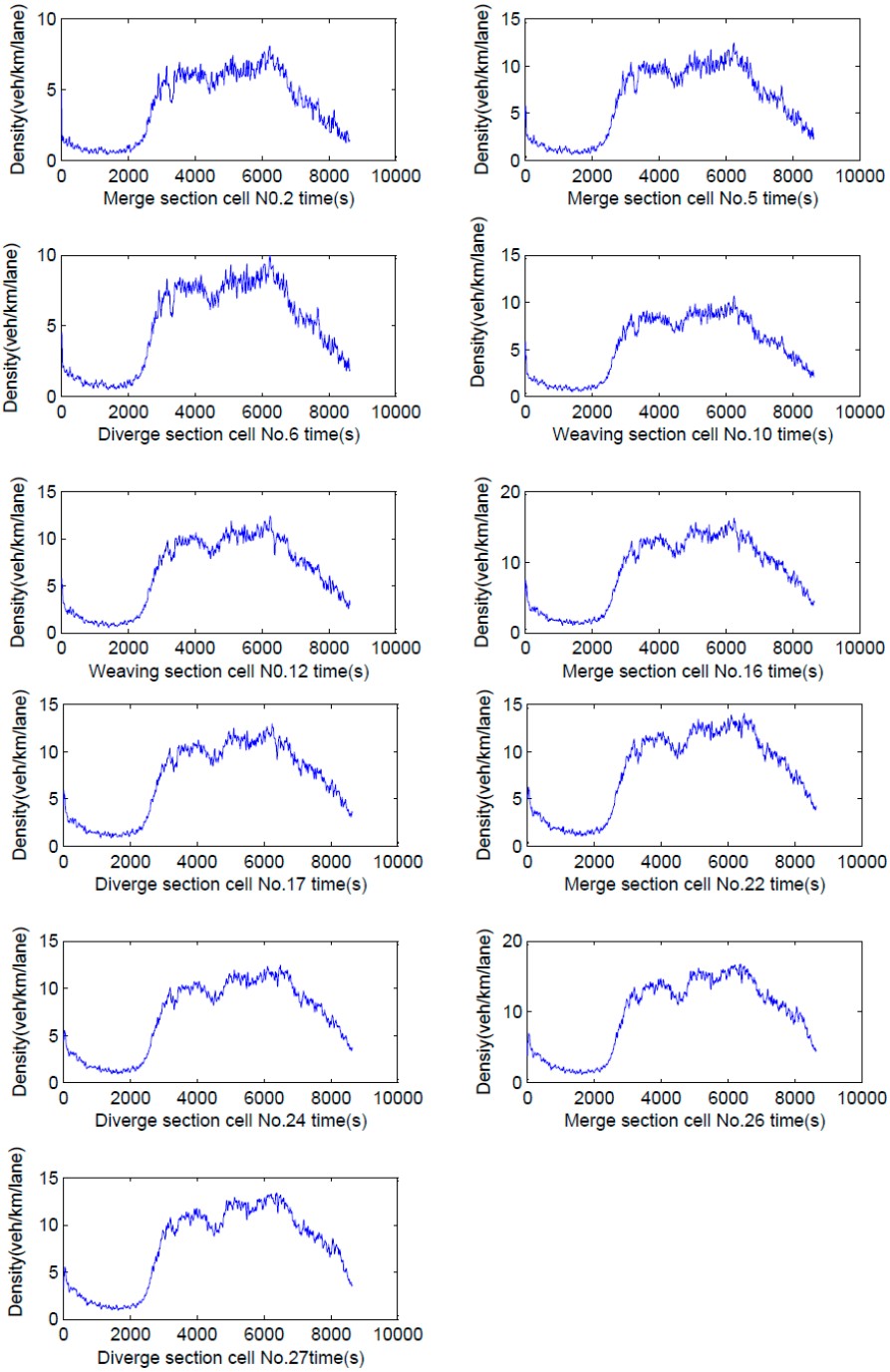

( b ) Diagram of merge/diverge/weaving section density on Non-weekdays

**Figure 7.** Diagram of merge/diverge/weaving section density evolution.

A diagram of the expressway traffic flow spatiotemporal evolution was presented in Figure 9. From Figure 9a, on weekday, the first congestion occurs on the expressway at around 07:30, and congested cell numbers include 6, 16, 24, and 26, which represent the diverge section and merge section. From Figure 9b, on the weekend, traffic congestion starts at the diverge section and merge section, the same as weekday. Comparing the spatiotemporal evolution graph of weekday and weekend, we can find congestion duration on weekend is longer than weekdays.

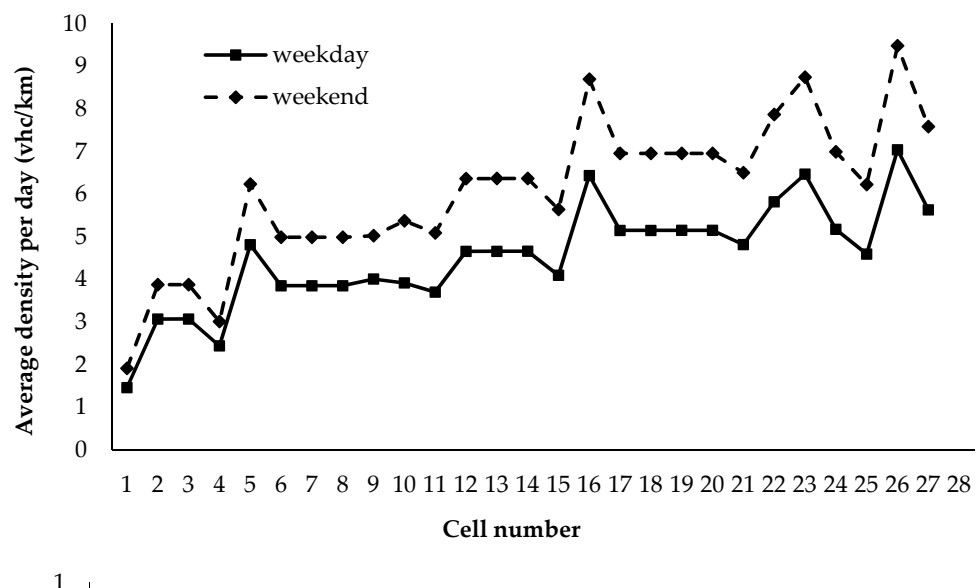

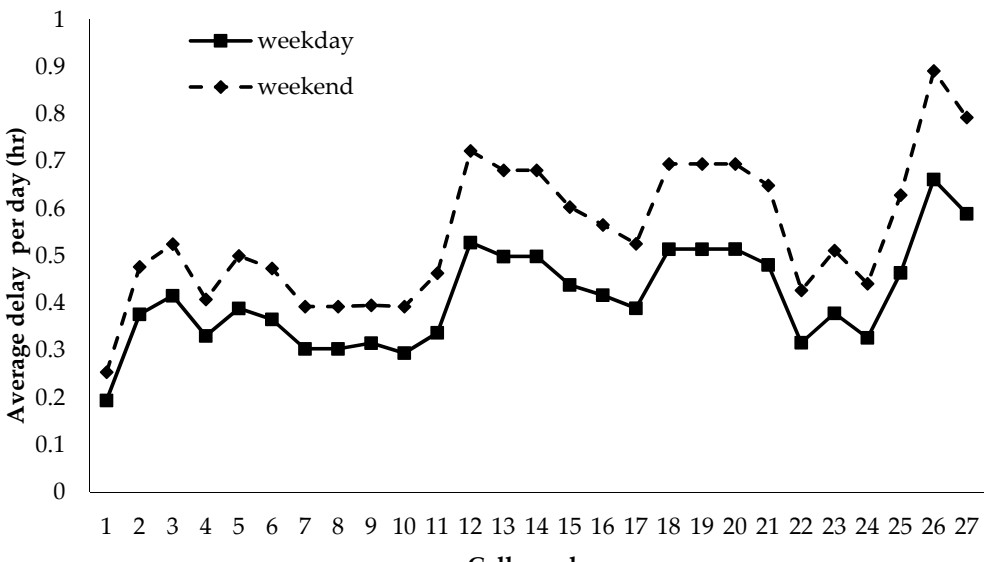

**Figure 8.** Average density/delay distribution at expressway.

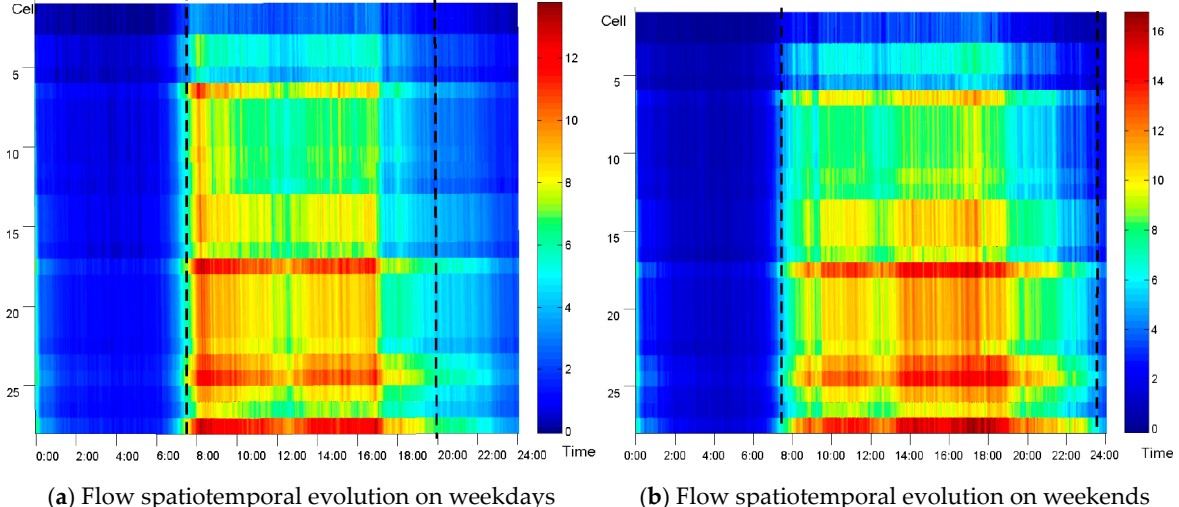

(**a**) Flow spatiotemporal evolution on weekdays (**b**) Flow spatiotemporal evolution on weekends

**Figure 9.** Diagram of the expressway traffic flow spatiotemporal evolution.

### 5.2. Sensitivity Analysis & Discussion

This section will discuss the influencing factors of traffic congestion in urban expressway mainlines, as follows.

#### 5.2.1. Effect of On-Ramp Traffic Volume on Mainline Traffic Congestion

There are seven on-ramps along Wanjiali Elevated Expressway. We will discuss the relation between on-ramp traffic volume and expressway mainline congestion. We collected traffic flow data from 10:00 to 11:00, 16 December 2017, and then the simulation was conducted under different on-ramp traffic flow input conditions. To simplify the drivers' unreasonable lane changing behavior, we define the merge rate $r_R$ as the ratio of the number of on-ramp vehicles to the number of vehicles on mainline of expressway. Four scenarios of different merge rate $r_R$ values were studied, in which $r_R$ = 0.2, 0.3, 0.4, and 0.5, in each.

Traffic delay of the merge area (Cell 5 in Figure 5) and weaving area (Cell 10 in Figure 5) were collected under different merge rate, $r_R$, illustrated in Figure 10.

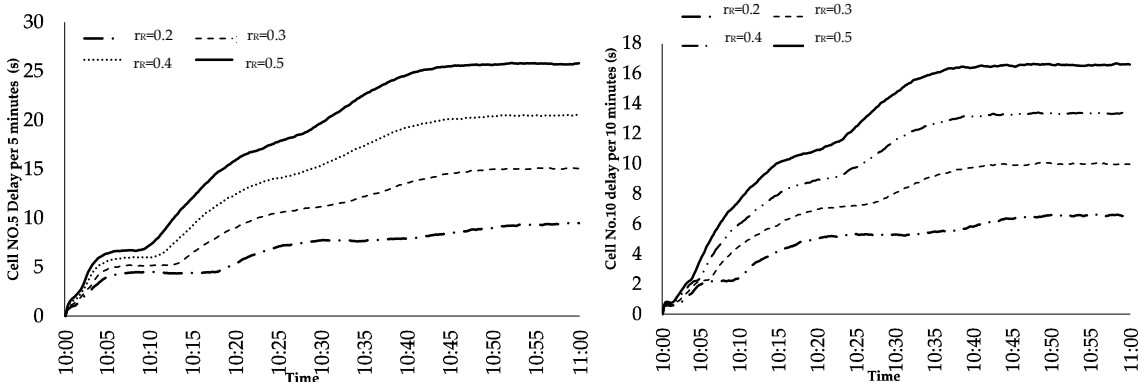

**Figure 10.** Merge section delay to on-ramp merge rate $r_R$.

From Figure 10, delay of the merge and weaving section increases with merge rate $r_R$, in addition, the increasing rate of delay decreases gradually. Further analysis can reveal the concrete effect of on-ramp traffic volume. When the merge rate $r_R$ increases from 0.2 to 0.3, the delay in the merge area (Cell 5) and weaving area (Cell 10) increases by 35% and 34%, respectively. When the merge rate, $r_R$, increases from 0.3 to 0.4, the delay in the merge area (Cell 5) and weaving area (Cell 10) increases by 26% and 25%. This phenomenon reveals that when merge rate $r_R$ increases, more vehicles will enter the merge/weave section, and interact with expressway mainline vehicles, and more disturbance will be generated in the traffic flow, due to the numerous unreasonable forced merging vehicles from on-ramp, and congestion will generate more frequently.

#### 5.2.2. Effect of Off-Ramp Capacity on Mainline Traffic Congestion

Off-ramp connects urban expressway mainline to the adjacent local arterial or street. Due to the short length of off-ramp, expressway traffic has a strong correlation with the local street. Therefore, capacity of the off-ramp $C_G$ is mostly determined by local streets. To this end, we conducted simulation under different off-ramp capacity $C_G$ conditions. Six scenarios of different off-ramp capacity were studied, in which $C_G$ = 500, 600, 700, 800, 900, and 1000 veh/hr, in each. Traffic data were collected from 10:00 to 11:00, on 16 December 2017, and applied for simulation. Delay in four diverge cells (No. 6, 17, 24, 27) was collected from simulation, which are shown in the Figure 11.

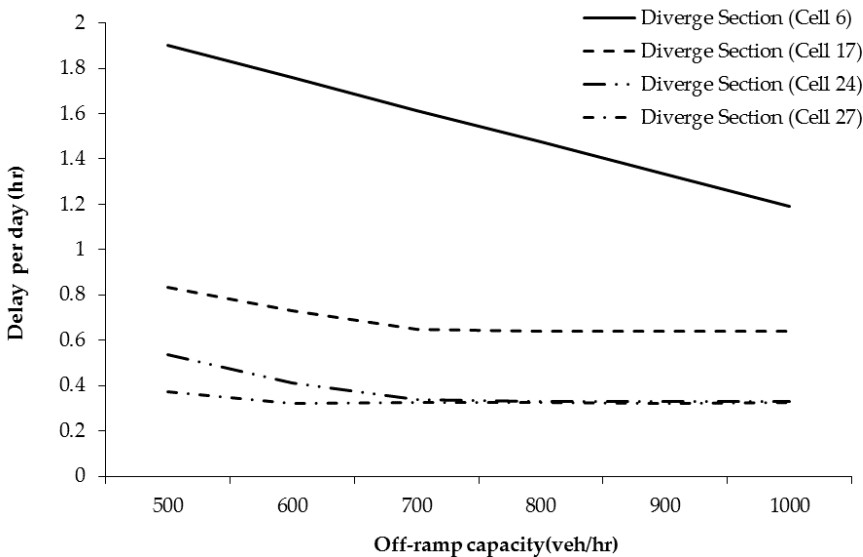

**Figure 11.** Diverge section delay to off-ramp capacity.

Figure 11 indicates that delay of diverge cell in the expressway mainline decreases with off-ramp capacity $C_G$ increasing. For diverge Cell 6, when $C_G$ increases by 100 veh/hr, the delay of cell 6 decreases by about 10%. For Cell 17 and Cell 24, in the former part, the diverge cell delay decreases with $C_G$ increasing and, in the latter part, when $C_G$ reaches 700 veh/hr, the delay of the diverge cell is basically unchanged. Similarly, when $C_G$ reaches 600 veh/hr, delay of the diverge for Cell 27 remains unchanged. To sum up, the capacity of local street will determine capacity of off-ramp, which has a strong effect on operating state at expressway mainline. As a result, traffic congestion occurring in the local road network will quickly spread to the expressway mainline, and even cause congestion.

*5.3. Discussion*

Comparing with traditional CTM, this paper proposed an improved CTM by introducing several parameters, including cell length and merge ratio. Considering the drivers' unreasonable behaviors, the improved CTM can accurately describe the traffic conditions of Chinese urban expressway. The method proposed in this paper was suitable for these scenarios:

- Typical urban expressway or freeway traffic characteristics in China. Due to the common phenomena of sudden lane changing on urban road, merge rates were applied in the improved CTM in conformity with changing lane behaviors on an urban expressway or freeway.
- Exploring the design and layout methods for ramps on urban expressway or freeway. This paper provided theoretical basis for verifying the rationality of the ramp design and analyzing the traffic capacity.
- Making freeway or expressway traffic control and management methods. This paper proposed an appropriate simulative method to traffic control and management on the urban expressway or freeway.

## 6. Conclusions

To analyze the mechanism of traffic congestion generation on the urban expressway in China, the Wanjiali Expressway, one representative urban elevated expressway in Changsha Hunan was selected as the study object. A cell transmission model, CTM, was applied to model and simulate the expressway traffic operation. Considering the particular characteristics of traffic operating on an urban expressway in China, CTM was improved, in which the cell length is variable, and microscopic drivers' mandatory merge behavior is integrated to a merge cell, and the limit of local streets capacity is linked to the diverge cell. The simulation was conducted based on Wanjiali Expressway conditions by the

improved CTM. Sensitivity analysis was conducted under different on-ramp volume and off-ramp capacity, and results indicate the specific characteristics and mechanism of traffic congestion at an urban expressway, which include:

- Most traffic congestion generates originally at merge, diverge, and weaving sections, then propagates to the next section upstream.
- Merge section congestion in an urban expressway is mostly caused by unreasonable driving behaviors, such as mandatory merging and lane-changing from on-ramp. Due to vehicles from on-ramp not obeying the "mainline priority" rule, on-ramp vehicles entering the merge section cell occupy a considerable ratio of the mainline capacity (sending flow), and the merge rate was used to represent the phenomena, and the delay of the merge and weaving sections increase between 25%–35% with the merge rate changed in the range of 0.2–0.4.

Diverge section congestion is mainly determined by the capacity of the local street connecting to expressway mainline through off-ramp. Based on the data analysis, delay of diverge cell in the expressway mainline decreases, at most, about 10%, with off-ramp capacity $C_G$ increasing per 100 veh/hr. Due to the unmatched capacity of off-ramp, the vehicle queue length often exceeds the off-ramp, and extends the mainline upstream. As a result, in the vehicle queue end, a dynamic bottleneck is generated at the diverge section, which reduces the capacity (receiving flow) of the diverge section cell.

Further research will be aimed at studying the congestion propagating feature on the urban expressway and, also, more study examples will be included to extend the methodology. In addition, CTM is a first order macroscopic model. On the one hand, compared with first order model, second order models can provide more realism. On the other hand, we considered changing lane behaviors on the expressway; nevertheless, this paper set the merge ratio to a constant value, which is different from the actual traffic conditions. Therefore, a second order model can be applied to this issue in future.

**Author Contributions:** This work was conducted by K.L., J.G. and W.W. with the help of graduate student Q.L. It was mainly drafted by K.L. and Q.L., and checked and revised by W.W. and J.G. K.L. and Q.L. designed and analyzed the proposed model. Q.L. and J.G. performed the simulation. L.D.H. and K.L. are responsible for the overall manuscript framework and language.

**Funding:** This research was funded by National Natural Science Foundation of China (NSFC; Grant No. 51678076), Hunan Key Laboratory of Smart Roadway and Cooperative Vehicle–Infrastructure Systems (Grant No. 2017TP1016), and Open Fund of Engineering Research Center of Catastrophic Prophylaxis and Treatment of Road & Traffic Safety of Ministry of Education (Grant No. KFJ170403) (Changsha University of Science & Technology).

**Acknowledgments:** The authors would like to give great thank to the hard work by the peer reviewers and editor. Also, we acknowledge the financial support provided by National Natural Science Foundation of China (NSFC; Grant No. 51678076), Hunan Provincial Key Laboratory of Smart Roadway and Cooperative Vehicle-Infrastructure Systems (Grant No. 2017TP1016), and Open Fund of Engineering Research Center of Catastrophic Prophylaxis and Treatment of Road & Traffic Safety of Ministry of Education (Grant No. KFJ170403) (Changsha University of Science & Technology).

**Conflicts of Interest:** The authors declare no conflict of interest.

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
