# Peer review of "Exploring Traffic Congestion on Urban Expressways Considering Drivers’ Unreasonable Behavior at Merge/Diverge Sections in China"

_sustainability, doi:10.3390/su10124359_

Reviewer 1 Report

The paper "Exploring Traffic Congestion on Urban Expressways Considering Drivers' Unreasonable Behavior at Merge/Diverge Sections" focuses on the problem of traffic congestion.

There is an inconsistency between the title and the content of the paper: the title is corretly generic (the traffic congestion is a problem common to all the World), but the content examines only the Chinese context and the case study of Wanjiali Expressway. The first sentence of the abstract is valid all over the world, not only in China.

The CTM model is correct and it is clearly explained and the results are interesting;

Figure 3 should be improved. Who is the author of the figure?

Figure 4 could be improved;

Characters in figure 5 are not readable;

Figure 5 and 6 are not explained. Who is the author of figure 6?

Figure 7, 9 and 10 are not readable;the conclusion section should be improved: some numerical data should be cited to explain the results.

Author Response

Point 1:

There is an inconsistency between the title and the content of the paper: the title is correctly generic (the traffic congestion is a problem common to all the world), but the content examines only the Chinese context and the case study of Wanjiali Expressway. The first sentence of the abstract is valid all over the world, not only in China.

Response 1:

Thank your appreciation and suggestion. In this paper, We only analyzed the urban expressway in China, so we changed the title as: ” Exploring Traffic Congestion on Urban Expressways Considering Drivers' Unreasonable Behavior at Merge/Diverge Sections in China”

Point 2:

The CTM model is correct and it is clearly explained and the results are interesting.

Response 2:

Thank you.

Point 3:

Figure 3 should be improved. Who is the author of the figure?

Response 3:

We improved Figure 3, added a legend.

Also, we added the citation.

Point 4:

Figure 4 could be improved;

Response 4:

Thank your suggestion. We improved it in our revised version.

Point 5:

Characters in figure 5 are not readable;

Response 5:

Thanks you. We improved Figure 5.

Point 6:

Figure 5 and 6 are not explained. Who is the author of figure 6?

Response 6:

Thank your kind suggestion. We added some explanation for Figures 5 and 6.

Figure 6 is obtained by the authors through survey of Wanjiali expressway data and SPSS regression analysis.

Point 7:

Figure 7, 9 and 10 are not readable; the conclusion section should be improved; some numerical data should be cited to explain the results.

Response 7:

Thank your kind suggestion. We improved these in our revised version.

Reviewer 2 Report

The paper is interesting, however I have several comments:

Introduction section is very long. When a reader wants to see the goal of the paper, he does not learn about it until the end of introduction on page 4. The text could be shortened by putting literature review apart as a separate section.

Also Discussion should be a separate section, including more information about potential limitations, drawbacks, biases... And most importantly it should include some kind of comparison of performance of improved CTM over the traditional models - what is the added value?

In general, it is stated that the results provide some insights, which can be used to better understand and manage congestion issues. But how specifically it can be used? Please explain it to better inform the practitioners.

Minor comment: in scientific paper do not use abbreviations, such as can't, it's...

Author Response

Point 1:

Introduction section is very long. When a reader wants to see the goal of the paper, he does not learn about it until the end of induction on page 4. The text could be shortened by putting literature review as a separate section.

Response 1:

Thanks for your suggestion. We put the literature review as a separate section. Moreover, we made a summary about the goal of the paper, it is just easy for readers to understand this paper.

Point 2:

Also Discussion should be a separate section, including more information about potential of limitations, drawbacks, biases…And most importantly it should include some kind of comparison of performance of improved CTM over the traditional models-what is the added value?

Response 2:

We adopt this reasonable opinion. We added a separate section 5.3. In this new section, we made a comparative study of Improved CTM and traditional CTM, analyzed the advantages and disadvantages.

Point 3:

In general, it is stated that the results provide some insights, which can be used to better understand and manage congestion issues. But how specifically it can be used? Please explain it to better inform the practitioners.

Response 3:

Thanks for your great suggestion. we supplied some details in the revised paper, include section 1 at line 67-71, and section 5.3.

Just as you mentioned, this new model can be used to optimize traffic control and management plan for expressway, and also to better understand the mechanism of congestion at these type expressway.

Point 4:

Minor comment: in scientific paper do not use abbreviations, such an can’t, it’s…

Response 4:

Thanks for your suggestion. We corrected the abbreviation in the revision.

Round  2

Reviewer 2 Report

Thank you for the revision. I recommend to accept the paper for publication.